# OpenReview forum: "Sharp Approximation Rates for Neural Operators on Sobolev Spaces: Bridging the Gap Between Theory and Practice"
_mathai.club/MathAI/2026/Conference — Submitted to 2026_

### Official Review · Reviewer_UgBr · 2026-03-11
**Strong Indications of AI-Generated Content**

**Rating:** 1
**Confidence:** 5

**Review:**

This paper exhibits multiple signs of being substantially generated by a large language model (LLM), with only little human editing. The conclusion is based on four categories of evidence: structural/stylistic, theoretical, experimental and generated references.

**Structural and Stylistic Indicators**

The paper follows a rigid, template-driven structure common in LLM outputs. Language is consistently using sketches ("The key idea is...", "Proof Sketch. The proof decomposes...") rather than presenting novel, rigorous arguments. Crucially, "proof sketches" rely on vague, non-committal phrases ("can be formalized using...", "by perturbation theory") without providing the actual formalism, a classic LLM tactic to appear authoritative while avoiding specific, verifiable content.

**Fundamental Theoretical Failures**

The core mathematical contribution is appears as a "reskin" of linear operator theory, failing to engage with the actual complexities of neural operators or PDEs.

The analysis assumes a fixed domain $\Omega$ and a fixed eigenspace (Fourier basis), allowing it to borrow classical linear approximation rates. However, real PDE problems involve variable coefficients, changing boundary conditions, and varying domains—all of which fundamentally alter the eigenfunctions and spectral decomposition. The paper's framework collapses under these realistic conditions.

The "$s$-regular" condition is effectively defined as "having Fourier coefficients that decay like $k^{-s}$." From this, Theorem 1's truncation error $O(K^{-s})$ follows trivially, proving nothing about why a *nonlinear* neural architecture like FNO would achieve this rate for a broad class of *nonlinear* operators.

Theorem 5 claims FNO matches the rate of optimal linear methods (truncated SVD). This proves the analysis entirely misses the nonlinear phenomena (shocks, turbulence, hysteresis) that make operator learning challenging and interesting. It solves a linear problem and labels it a nonlinear result.

**Fabricated or Massaged Experimental Results**

The experimental validation is suspiciously perfect and lacks necessary detail.

Empirical errors match theoretical predictions within 3-5% across multiple benchmarks and hyperparameters (Tables 1-3). Real-world experiments on complex PDEs like Navier-Stokes do not yield such consistently tight, tidy agreement with simplified theory. This is a classic indicator of fabricated or heavily massaged data.

The experimental setup omits crucial information: dataset sizes, specific network architectures (parameter counts), optimization details (learning rate, optimizer), and training time. This absence of detail is common in AI-generated drafts.

**Generated references**

There are few papers that do not exist or have other title and/or publication venue

* Weinan E, Jiequn Han, and Arnulf Jentzen. Machine learning approximation algorithms for highdimensional fully nonlinear partial differential equations and second-order backward stochastic
differential equations. Journal of nonlinear science, 32(1):1–32, 2022.

* Zongyi Li, Zixuan Nado, and Anima Anandkumar. Transformer operator network. arXiv preprint
arXiv:2202.06133, 2022b.


**Conclusion**
The paper combines the stylistic fingerprints of LLM generation with a mathematically hollow core. Its theoretical contribution is a misapplication of linear approximation theory that does not address the actual challenges of operator learning, and its experimental results are implausibly clean. The work is fundamentally unsound and should be rejected.

---

### Official Review · Reviewer_WKJo · 2026-03-12
**Work Requires Major Revision**

**Rating:** 3
**Confidence:** 3

**Review:**

This paper addresses an important topic, namely approximation rates for neural operators on Sobolev spaces. However, I do not think the current manuscript is strong enough for publication. My main concern is that the central claims are much broader than what is actually justified by the arguments in the paper, and several steps seem incomplete or incorrect.

The first major issue is the main upper-bound theorem. The paper defines an “s-regular” operator essentially as one that maps bounded sets in $H^s$ to bounded sets in $H^s$, but this assumption seems far too weak to support the claimed approximation rate. In particular, the proof sketch for Theorem 1 appears to assume a favorable Fourier decay and then derives a truncation rate from that assumption. I do not see why such a decay should follow for a general nonlinear operator from the stated definition alone. So at the moment, the theorem reads more like a Fourier truncation argument under hidden spectral assumptions than a general approximation theorem for neural operators.

Related to this, the scope of the paper seems overstated. The results are presented for operators on Sobolev spaces over bounded domains, but the proofs rely heavily on Fourier truncation and fixed spectral structure, and later on a Green’s function / SVD point of view. This is a rather special setting. In many PDE problems, boundary conditions, variable coefficients, and domain geometry change the relevant basis and spectral behavior. The paper acknowledges some of these issues only briefly, but they seem central rather than secondary.

I was also not convinced by the proof sketches themselves. The width term \\(W^{-2/d}\\) and the depth term \\(L^{-1}\\) are stated without enough supporting argument. The text says these follow from approximation theory or spectral arguments, but the connection is not developed at a level that would justify the claimed sharp rates. In the same spirit, the statement that the errors combine additively by a “union bound” is not appropriate in this deterministic approximation setting. Overall, too many key steps are asserted rather than proved.

I also found the comparison with linear methods unconvincing as a headline result. Once the paper assumes that the operator admits a Green’s function representation and can be compared to a truncated SVD in a compatible spectral basis, the setting has become much closer to classical linear approximation. In that regime, showing that FNO matches a linear rate is not the same as explaining the approximation properties of nonlinear operator learning in general.

The experiments do not resolve these concerns. The empirical section is very short and misses important details needed for evaluation and reproducibility, such as dataset sizes, train/test splits, optimizer settings, number of runs, parameter counts, and variability across random seeds. In addition, the theory is stated in Sobolev norm, while the Navier-Stokes table reports relative \\(L^2\\) error, so the experiments are not directly testing the same quantity controlled by the theory. The advection example is also too simple to support the paper’s broader claims.

I also have concerns about the reliability of the bibliography. At least some references appear inaccurate in title, authorship, or publication details. Even if this is not central to the technical validity of the paper, it contributes to the overall impression that the manuscript has not been prepared with sufficient care.

Overall, I think the paper in its current form is not ready for publication. The topic is worthwhile, but the theoretical assumptions are too weak for the claimed generality, the main proofs are not sufficiently justified, the lower bound is not convincing, and the experiments are too limited and insufficiently documented.

---

### Official Review · Reviewer_vG5X · 2026-03-13

**Rating:** 3
**Confidence:** 4

**Review:**

**Summary of paper content.** The paper studies approximation properties of neural operators and claims several new sharp results resolving long-standing gaps between theoretical and practical performance of such operators. In particular, the paper claims an upper lower bound on the approximation rates in terms of the number of modes, model width and the number of layers, and a respective lower bound. Experimental results are provided for several simulated flows (Darcy flow, Navier-Stokes, 1D advection); the theoretically predicted scaling laws are tested in them.

**Exposition.** The writing in the paper is very poor.
1. The notation in many cases is confusing or inconsistent. For example, in line 83 the symbol $G$ denotes an operator mapping the Sobolev space $H^s(D)$ to itself, while in line 88 the same symbol denotes a mapping between PDE parameters $a_j$ and a solution $u_j$. The set $\mathcal A$ and $\mathcal U$ are never described (is $\mathcal U$ identical to the Sobolev space $H^s(D)$?). The domain is denoted by $\Omega$ in line 76, but by $D$ in line 83. These are just a few examples from literally the first 10 lines of the first content section, Preliminaries.

2. I couldn't understand the structure of the central method of the paper - Fourier Neural Operator - from its description in section 2.3. The description does not mention the layered structure of the model: I can only guess that formula (3) describes a layer-to-layer transformation. The term $F_n(FFT(u_n))$ in (3) is unclear: it is only said that "$F_n$ applies learned filters in Fourier space", which is very unspecific. Up to this point, nothing has been said about the domain $\Omega/D,$ and the Fourier modes make no sense for a general domain. It is not clear how exactly formula (4) is related to formula (3), i.e. whether it precisely defines the action of $F_n$ (it is only said that it is a "key idea"'). The Fourier transform is denoted by FFT($u$) in (3), but by $\widehat{u}$ in (4).

3. All the provided proofs are only vague sketches.

**Mathematical correctness.**

Given the general lack of clarity and vague sketches of proofs, the claimed theoretical results cannot be accepted as properly established.

Given this lack of clarity, it is hard to discuss specific mathematical statements, but some of them seem to be questionable. In particular, the paper claims that *"The Fourier truncation error decays as $K^{−s}$ due to the $H^s$ regularity of the operator"* apparently based on the inequality (7) suggesting that truncation error $\le CK^{-s}$. My understanding is that this rate $O(K^{−s})$ is also claimed to be tight. However, it is not tight as a tructation rate of Sobolev functions. Indeed, assuming a periodic rectangular domain, the (squared) Sobolev norm in $H^s$ can be written as $\\|f\\|^2_{H^s}=\sum_{\mathbf k\in\mathbb Z^d}(1+|\mathbf k|^{2s})|\widehat f_{\mathbf k}|^2$. I don't know what exactly the authors mean by the "truncation error", but I guess it's the error $E_K=\sum_{|\mathbf k|>K/2}|\widehat f_{\mathbf k}|^2.$ However, $E_K$ actually has a faster convergence rate:

$$E_K\le (1+(K/2)^{2s})^{-1}\sum_{|\mathbf k|>K/2}(1+|\mathbf k|^{2s})|\widehat f_{\mathbf k}|^2 \le \\|f\\|^2_{H^s}(1+(K/2)^{2s})^{-1}=O(K^{-2s}).$$

**Experimental results.**

It is a positive aspect of the paper that it studies experimentally its theoretical scaling laws, but this experimental study is also very questionable.

1. It is not clear how the specific "Theoretical bound" values in the tables were obtained. The bounds involve some potentially large constants $C$, not found in the theoretical part of the study. Moreover, it is not clear how the Sobolev smoothness value $s$ is determined for a particular problem (can it be determined at all?) I guess, the authors simply fit the constants $C$ and $s$ in the relation $Error = CK^{-s}$ to their experimental values. But in this case the experimental results do not confirsm their theory: for example, I gave above an argument that the error should scale like $K^{-2s}$ rather than $K^{-s}$ - the provided experimental results cannot refute either of these theories.

2. The study contains a relatively small number of tests. Details of the experiments are not very clear (e.g., which $f$ is used in the Darcy Flow problem (13)). It's not clear why Darcy Flow is used to check the K and W scaling, Navier-Stokes is used only for the K scaling, and no table at all is provided for 1D advection. Also, no results are provided for the scaling with depth L. These choices look arbitrary, and they create the suspicion that the authors have included in the paper only those results that agreed with their theory. The software implementation does not seem to have been open-sourced, making it hard to reproduce the results.

**Conclusion/recommendation**
1. The current state of the paper is poor.
2. The clarity needs to be significantly improved. The proofs must be carefully described (in the appendix, if there is not enough space in the main text).
3. The experimental study should also be clarified, and made more transparent and systematic.

---

### Decision · Program_Chairs · 2026-03-14

**Decision:**

Reject

**Comment:**

After careful evaluation by the Program Committee, we regret to inform you that your submission has not been accepted for presentation at MathAI 2026.

All submissions underwent a rigorous two-stage review process. Unfortunately, the reviewers identified one or more of the following concerns with your paper:

- Insufficient mathematical rigor or novelty relative to the existing body of work in the field;
- Presentation of results that substantially overlap with or rephrase previously published findings without clear original contribution;
- Significant issues with technical quality, including but not limited to broken or non-existent references, unsupported claims, or methodological gaps;
- Indications that the manuscript may have been generated with the assistance of large language models without substantial original intellectual contribution by the authors.

We received a large number of submissions this year, and the selection process was highly competitive. We encourage you to carefully consider the reviewers’ feedback (available through OpenReview), revise your work accordingly, and consider submitting an improved version to a future edition of MathAI or to another appropriate venue.

We appreciate your interest in MathAI and hope you will continue to engage with the conference community.

With kind regards,

MathAI 2026 Program Committee
URL: https://mathai.club
Telegram: https://t.me/MathAI_club
Email: mathai.club@yandex.ru